# Maximizing Antenna Array Aperture Efficiency for Footprint Patterns

**DOI:** 10.3390/s23104982

**Published:** 2023-05-22

**Authors:** Cibrán López-Álvarez, María Elena López-Martín, Juan Antonio Rodríguez-González, Francisco José Ares-Pena

**Affiliations:** 1Center for Research in NanoEngineering, Campus Diagonal-Besòs, Polytechnic University of Catalonia, 08019 Barcelona, Spain; cibran.lopez@upc.edu; 2Department of Morphological Sciences, University of Santiago de Compostela, 15782 Santiago de Compostela, Spain; melena.lopez.martin@usc.es; 3Department of Applied Physics, University of Santiago de Compostela, 15782 Santiago de Compostela, Spain; ja.rodriguez@usc.es

**Keywords:** antenna array apertures, pattern synthesis, footprint patterns

## Abstract

Despite playing a central role in antenna design, aperture efficiency is often disregarded. Consequently, the present study shows that maximizing the aperture efficiency reduces the required number of radiating elements, which leads to cheaper antennas with more directivity. For this, it is considered that the boundary of the antenna aperture has to be inversely proportional to the half-power beamwidth of the desired footprint for each ϕ-cut. As an example of application, it has been considered the rectangular footprint, for which a mathematical expression was deduced to calculate the aperture efficiency in terms of the beamwidth, synthesizing a rectangular footprint of a 2:1 aspect ratio by starting from a pure real flat-topped beam pattern. In addition, a more realistic pattern was studied, the asymmetric coverage defined by the European Telecommunications Satellite Organization, including the numerical computation of the contour of the resulting antenna and its aperture efficiency.

## 1. Introduction

In IEEE Standard for Definitions of Terms for Antennas [1], the antenna illumination efficiency is defined as the ratio of the maximum directivity of an antenna aperture to its standard directivity, and the antenna aperture efficiency as the ratio of the maximum effective area of the antenna to the aperture area. In some cases, illumination and aperture efficiencies might coincide. It is remarkable that those antenna arrays synthesized with small aperture efficiency require more radiating elements than necessary.

Kim and Elliott [2] proved that the extensions of Tseng–Cheng distributions, which give a flat-topped beam in every ϕ-cut, are inefficient because they use rectangular boundary arrays, and, as a consequence, the obtained shaped patterns are almost rotationally symmetric, and they present ring side lobes that are not circular ($\theta_{peak}\neq constant$). Obviously, the optimal boundary for this distribution should be circular and not rectangular.

The vast majority of works regarding footprint pattern synthesis use rectangular, circular, or elliptical antennas independently of the shape of the desired pattern ([3,4,5,6,7,8,9,10] being some representative examples). Even some works related to reflectarrays suffer from this issue [11,12].

In [13,14], a synthesis was implemented that tried to slightly optimize antenna aperture efficiency, but without analyzing the problem in depth.

Elliott and Stern [15] have suggested that, in order to obtain a highly efficient antenna, its contour has to be inversely proportional to its half-power beamwidth (HPBW) in every ϕ-cut. This technique was developed by Ares et al. [16] in order to synthesize square footprints. Afterwards, Fondevila et al. [17] numerically optimized the contour of an antenna to obtain rectangular footprints.

More recently, López-Álvarez et al. [18] presented an efficient iterative method that, starting from a circular aperture and removing those elements with low-amplitude excitations, generates footprint patterns.

In this work, a study is presented which tries to maximize aperture efficiency for rectangular footprints as well as for the case of the asymmetric coverage defined by the European Telecommunications Satellite Organization (EuTELSAT) for the first time to the best of our knowledge. The role of aperture efficiency in the synthesis of high-performance antennas is highlighted, a topic that is usually disregarded in modern studies, given that, as previously stated, high aperture efficiencies guarantee not using more radiating elements than needed. The use of conventional methods, which do not maximize aperture efficiency, would increase the price and even diminish the directivity for those antennas in which illumination and aperture efficiencies coincide, requiring larger antenna areas.

In order to synthesize optimal antenna patterns with specific ripple and side-lobe levels, it is necessary to optimize the disposition of radiating elements within the antenna. If this is not achieved, the antenna contour has to be optimized. This work proposes antenna contours that are very close to the optimal solution, which allows obtaining this with global or even local optimization methods.

## 2. Materials and Methods

A unidirectional planar, circular aperture of radius *a* and continuous aperture distribution $K_{0}\left(\rho \right)$ (that is, a linearly polarized planar aperture distribution), with the notation expressed in Figure 1, produces the ϕ-symmetric pattern [19,20] from Equation (Equation 1).
(1)$$F\left(\theta \right)=2\pi \int_{0}^{a}K_{0}\left(\rho \right)J_{0}\left(k\rho \sin\left(\theta \right)\right)\rho d\rho$$
where λ is the wavelength, $J_{0}$ is the zeroth Bessel function of the first kind, and θ is the azimuthal angle, that is, the angle measured from the zenith of this aperture. Consider the following relations:(2)$$u=\frac{2a}{\lambda }\sin\left(\theta \right);\;p=\frac{\pi }{a}\rho ;\;g_{0}=\frac{2a^{2}}{\pi }K_{0}\left(\rho \right)$$
where the parameter *u* defines the pointing direction in real space, consisting of an equation that relates the angle and the wavelength, this angle defining the HPBW of the flat-topped beam, and ρ is the radial coordinate of the aperture. These substitutions transform the last equation into
(3)$$F\left(u\right)=\int_{0}^{\pi }pg_{0}\left(p\right)J_{0}\left(up\right)dp$$

For instance, the case of a constant, uniform aperture $g_{0}\left(p\right)=1$ yields the well-known pattern
(4)$$F_{0}\left(u\right)=\frac{J_{1}\left(\pi u\right)}{\pi u}$$
consisting of a main beam surrounded by a family of ring side lobes (given the existing axial symmetry), where $J_{1}$ is the first-order Bessel function of the first kind.

Then, by representing the aperture distribution in terms of the roots of $J_{1}$, such that $J_{1}\left(\pi \gamma_{1n}\right)=0,n=0,1,2\ldots$, we obtain [19]:(5)$$g_{0}\left(p\right)=\sum_{n=0}^{\infty }B_{n}J_{0}\left(\gamma_{1n}p\right)$$

Integrating the initial ϕ-symmetric pattern (Equation (Equation 3)) with the previous aperture distribution (Equation (Equation 5)), evaluated at the roots $\gamma_{1n},n=0,1,2\ldots$, we obtain the aperture distribution
(6)$$g_{0}\left(p\right)=\frac{2}{\pi^{2}}\sum_{n=0}^{\infty }\frac{F\left(\gamma_{1n}\right)J_{0}\left(\gamma_{1n}p\right)}{J_{0}^{2}\left(\gamma_{1n}\pi \right)}$$

Thus, if the roots $u_{n}=\gamma_{1n}$ for $n\geq \bar{n}$ are kept ($\bar{n}$ being the transition parameter), but the inner roots are displaced for $n=1,2,\ldots ,\bar{n}-1$ to the new complex positions $u_{n}+jv_{n}\neq \gamma_{1n}$, the pattern becomes a rotationally symmetric field that can be radiated by a circular aperture, with properly filled nulls in the shaped region and controlled side lobe levels in the unshaped region. With appropriate values of $u_{n}$ and $v_{n}$, it is possible to synthesize both real and complex flat-topped beam patterns using [15,17,19]. This is accomplished by dividing Equation (Equation 4) by its (1+ϵ)M+s first zeros and multiplying by the new, displaced ones:(7)$$F\left(u\right)=\frac{J_{1}\left(\pi u\right)}{\pi u}\frac{\prod_{n=1}^{M}\left(1-\frac{u^{2}}{{\left(u_{n}+jv_{n}\right)}^{2}}\right){\left(1-\frac{u^{2}}{{\left(u_{n}-jv_{n}\right)}^{2}}\right)}^{\epsilon }\prod_{n=M+1}^{M+s}\left(1-\frac{u^{2}}{u_{n}^{2}}\right)}{\prod_{n=1}^{(1+\epsilon )M+s}\left(1-\frac{u^{2}}{\gamma_{1n}^{2}}\right)}$$
where ϵ={0,1} (the pattern is real if ϵ=1 and complex if ϵ=0). As a result, $F(u_{n})\neq 0$ unless $v_{n}=0$; thus, there exist new complex roots positions ($u_{n}+jv_{n}$) for which the pattern has properly filled roots in the shaped region and controlled side lobe levels in the rest. For n∈(M+1,M+s), the peak levels of the inner *s* side lobes depend on the values of $u_{n}$, with a decay of $u^{-3/2}$ for distant side lobes. The flat-topped beam is composed of a central beam surrounded by *M* annular ripples of the same height, with the depth of the troughs between these components depending on the $u_{n}$ and $v_{n}$ for n∈[1,M]. The corresponding aperture distribution given by Equation (Equation 6) truncates at $\bar{n}=\left(1+\epsilon \right)M+s+1$.

Therefore, a flat-topped beam extended about $u_{0}=\frac{2a}{\lambda }\sin\left(\theta_{0}\right)$ such that $F(u_{0})=-$3 dB will give a $\theta_{0}$ value that is smaller if $\frac{a}{\lambda }$ is larger. The achieved flat-topped beam for real patterns (ϵ=1) is broader than those corresponding to the complex patterns (ϵ=0), and the $u_{0}$ value is also bigger in real patterns. The angle $\theta_{0}$ will define the HPBW of the flat-topped beam. Consequently, the flat-topped HPBW is inversely proportional to $\rho_{max}\left(\beta \right)$, which is the distance along the β line in the XY plane out of the periphery.
(8)$$\frac{2a}{\lambda }\sin\left(\theta_{0}\right)=\frac{2\rho_{max}\left(\beta \right)}{\lambda }\sin\left(\theta (\beta )\right)$$

Thus, the product of the antenna size by the HPBW ($a\cdot \beta \omega_{0}$) is conserved:(9)$$a\cdot \beta \omega_{0}=\rho_{max}\left(\beta \right)\cdot \beta \omega \left(\beta \right)$$

As a particular case, we can now consider a rectangular footprint, with quadrant symmetry. The maximum radius for the rectangular footprint is
(10)$$\rho_{max}\left(\beta \right)=\left\{\begin{array}{c} \frac{\beta \omega_{0}}{\beta \omega_{a}}a\cdot \cos\left(\beta \right)\mathrm{if}0\leq \beta <\alpha \\ \frac{\beta \omega_{0}}{\beta \omega_{b}}a\cdot \sin\left(\beta \right)\mathrm{if}\alpha \leq \beta <\pi /2 \end{array}\right.$$

Given this, for some angle α, $\rho_{max}\left(\beta \right)$ must be the same for both cases, with $\beta \omega_{a},\beta \omega_{b}$ being the HPBW in each axial direction:(11)$$\tan\left(\alpha \right)=\frac{\beta \omega_{b}}{\beta \omega_{a}}$$

By considering $\beta \omega_{a}\geq \beta \omega_{b}$, and given that the required antenna for synthesizing this specific pattern (from Equation (Equation 10)) cannot exceed the available one (that is, the antenna defined initially, a circular aperture of radius *a*), it can be shown that there is the following constraint regarding the ratios of the HPBW in each axial direction:(12)$$\begin{array}{c} \frac{\beta \omega_{0}}{\beta \omega_{b}}a\sin\beta \cos\beta =\frac{\beta \omega_{0}}{\beta \omega_{b}}a\frac{\sin2\beta }{2}=f\left(\beta \right)\leq a \end{array}$$
with $\alpha \leq \beta \leq \frac{\pi }{2}$. It is straightforward to obtain the maximum value of the function f(β):(13)$$\begin{array}{c} \frac{df}{d\beta }=0\Rightarrow \frac{\beta \omega_{0}}{\beta \omega_{b}}a\cos2\beta =0\Rightarrow \beta_{m}=\frac{\pi }{4} \end{array}$$

For that angle, the function takes the value
(14)$$\begin{array}{c} f\left(\beta_{m}\right)=a\frac{\beta \omega_{0}}{\beta \omega_{b}}\frac{1}{2} \end{array}$$

Considering that the HPBW associated with the aperture of radius *a* is such that $\beta \omega_{0}=\beta \omega_{a}$, the constraint regarding the ratios of the HPBW results in
(15)$$\begin{array}{c} f\left(\beta_{m}\right)=a\frac{\beta \omega_{a}}{\beta \omega_{b}}\frac{1}{2}\Rightarrow \beta \omega_{a}\leq 2\beta \omega_{b} \end{array}$$
therefore:(16)$$\begin{array}{c} 1\leq \frac{\beta \omega_{a}}{\beta \omega_{b}}\leq 2 \end{array}$$
or, otherwise, the shape of the antenna could not verify the aspect ratio for the desired footprint.

The effective area of the antenna, in radial coordinates, is computed as the sum of two terms, depending on the value of $\rho_{max}\left(\beta \right)$:(17)$$A_{e}=\frac{1}{2}\int_{0}^{\frac{\pi }{2}}{\rho^{2}}_{max}\left(\beta \right)d\beta$$

Therefore, the effective area (for a quadrant) is
(18)$$A_{e}=\frac{{\left(a\cdot \beta \omega_{0}\right)}^{2}}{8}\left(\frac{2\alpha +\sin\left(2\alpha \right)}{{\left(\beta \omega_{a}\right)}^{2}}+\frac{\pi -2\alpha +\sin\left(2\alpha \right)}{{\left(\beta \omega_{b}\right)}^{2}}\right)$$

On the other hand, the antenna aperture area for such a pattern (with the shape of a rectangle) is
(19)$$A=\frac{{\left(a\cdot \beta \omega_{0}\right)}^{2}}{\left(\beta \omega_{a}\right)\left(\beta \omega_{b}\right)}$$

The antenna efficiency, as previously defined, is
(20)$$\eta_{a}=\frac{A_{e}}{A}=\frac{1}{8}\left(\frac{\beta \omega_{b}}{\beta \omega_{a}}\left(2\alpha +\sin\left(2\alpha \right)\right)+\frac{\beta \omega_{a}}{\beta \omega_{b}}\left(\pi -2\alpha +\sin\left(2\alpha \right)\right)\right)$$

The effective area of the antenna can be expressed in terms of the directivity [20] as
(21)$$A_{e}=\eta_{a}A=D_{max}\frac{\lambda^{2}}{4\pi }$$
where *A* is the area of the antenna, and $\eta_{a}\leq 1$ is the antenna efficiency of an aperture-type antenna. On the other hand, the standard directivity [21], the directivity that can be obtained with an aperture *A*, is
(22)$$D_{std}\leq A\frac{4\pi }{\lambda^{2}}$$

As a result, considering the maximum value for $D_{std}$, $\eta_{i}$ being the illumination efficiency and taking into account Equation (Equation 21), we have
(23)$$\eta_{i}=\frac{D_{max}}{D_{std}}=\frac{\lambda^{2}}{4\pi }\frac{D_{max}}{A}=\frac{A_{e}}{D_{max}}\frac{D_{max}}{A}=\frac{A_{e}}{A}=\eta_{a}$$
that is, the illumination efficiency equals the antenna aperture efficiency. Thus, in this case, a good antenna aperture efficiency implies a good illumination efficiency.

## 3. Discussion

### 3.1. Application to Footprints with Quadrant Symmetry

For this set of applications, Equation (Equation 20) is implemented, taking into account quadrantal symmetry. We might check the case of a clover generating a square pattern (Figure 2). Considering $\alpha =\frac{\pi }{4}$ and $\beta \omega_{a}=\beta \omega_{b}$, the aperture efficiency would be
(24)$$\eta_{1:1}=0.64$$

As an example of application of the synthesis of a square footprint pattern of approximately $20^{\circ }\times 20^{\circ }$ using an aperture of the shape of Figure 2, in [16], a continuous aperture distribution g(p) (that is, pure real $g_{0}\left(p\right)$ from Equation (Equation 6) truncated at $\bar{n}=\left(1+\epsilon \right)M+s+1$ with ϵ=1, M=2, and s=2) was stretched into a distribution within its boundary and afterwards sampled to be applied to a rectangular grid array; 36% of the elements would be saved (thus, 52 elements of a total of 144 for each quadrant). This coincides with the fact that the effective area is 64% of the total area (from Equation (Equation 24)).

Then, a rectangular footprint pattern of a 2:1 aspect ratio (Figure 3), with $\tan\left(\alpha \right)=\frac{1}{2}$, leads to
(25)$$\eta_{2:1}=0.86$$

We consider a rectangular footprint pattern of a 3:1 aspect ratio (Figure 4), with $\tan\left(\alpha \right)=\frac{1}{3}$. As can be seen, the required antenna exceeds the available one, indicated with solid lines in Figure 4. Thus, the real aperture efficiency $\eta^{r}$ has to consider the complete required antenna, with dashed lines Figure 4, which are
(26)$$\begin{array}{c} \eta_{3:1}=1.21\\ \eta_{3:1}^{r}=0.80 \end{array}$$

Furthermore, this effect is accentuated for a rectangular footprint pattern of a 4:1 aspect ratio (Figure 5), with $\tan\left(\alpha \right)=\frac{1}{4}$, which leads to
(27)$$\begin{array}{c} \eta_{4:1}=1.59\\ \eta_{4:1}^{r}=0.80 \end{array}$$

The synthesis of a rectangular footprint of a 2:1 aspect ratio is exemplified, by starting from a pure real flat-topped beam pattern (ϵ=1) with a side-lobe level SLL=−25 dB, $\bar{n}=6$, M=2 filled nulls, and a ripple level of ±0.5 dB; the method depicted in [18] synthesized a pattern with SLL=−25 dB and a ripple level of ±0.8 dB. The resulting array has 1044 elements 0.5λ spaced. Figure 6 shows the normalized aperture distribution as well as the final pattern.

### 3.2. Application to Asymmetric Footprints

As an initial pattern to compose all the footprints in this section, both real and complex (obtained with the methods described in [15,19] and shown in Figure 7 and Figure 8, respectively) flat-topped beam pattern boundaries are considered. Both sets of roots are implemented with two filled zeros (Figure 9) and one only filled zero (Figure 10). The latter requires much smaller antennas than the former.

For this case, the contour of the antenna has to be numerically computed with Equation (Equation 9), the values of $u_{0}$ from the diagrams of Figure 7 and Figure 8, and the area of the antenna ($A_{e}$).

The European coverage defined by the EuTELSAT yields interesting applications from geostationary satellites, with the contour from Figure 11. This configuration leads to the same aperture efficiency for the real and complex cases ($\eta_{EuTELSAT}^{real}=\eta_{EuTELSAT}^{complex}=0.778$).

## 4. Conclusions

As a consequence of our study, it has been proved that, for rectangular footprints, the HPBW within the principal planes must verify $1\leq \frac{\beta \omega_{a}}{\beta \omega_{b}}\leq 2$, i.e., a maximum aspect ratio of 2:1, in order to be able to obtain an aperture size fitting the aspect ratio of the footprint.

For the case of the EuTELSAT antenna contour, which has been synthesized using both real and complex diagrams, it has been found that the antenna shape from the real pattern is always bigger than from the complex one, but maintaining the same aperture efficiency. If the number of ripple levels is reduced to one, the shape of the antenna also decreases for both pure real and complex cases, verifying the former result. The implementation in this case would lead to the use of fewer elements at the expense of also reducing the antenna directivity in comparison with the case of two ripple levels. For both real and complex patterns, reducing the ripple level implies a small shrinkage of the radiation pattern in the shaped region, which will be greater as the SLL increases.

In the examples shown, we have considered the most favorable case, as we have always used the minimum aperture area that adjusts the effective area as much as possible. Nevertheless, in real cases, the antenna aperture area is expected to be bigger.

This procedure is directly applicable to equispaced linear arrays, where the product of the number of elements and HPBW is approximately constant.

It is recommended that these possible improvements in the aperture efficiency be incorporated in all future syntheses of linear and planar arrays.

## Figures and Tables

**Figure 1 sensors-23-04982-f001:**
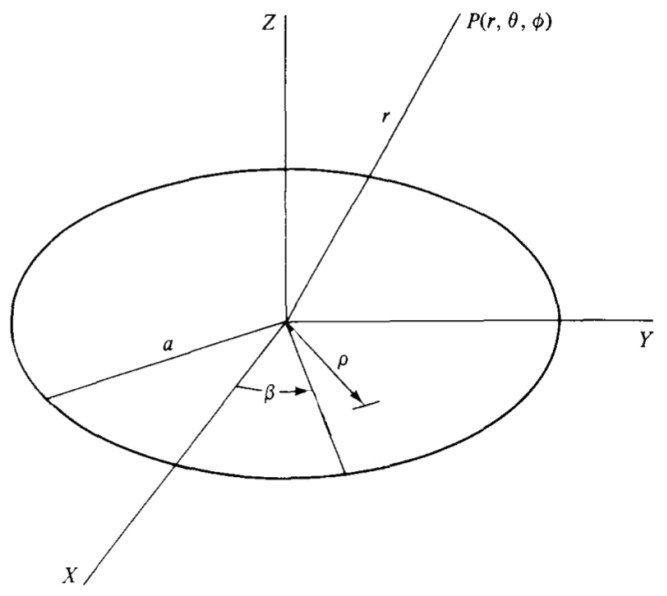
Circular aperture of radius *a*. The cylindrical coordinates ρ,β refer to the antenna aperture, while the spherical coordinates r,θ,ϕ are used for defining the field point *P*.

**Figure 2 sensors-23-04982-f002:**
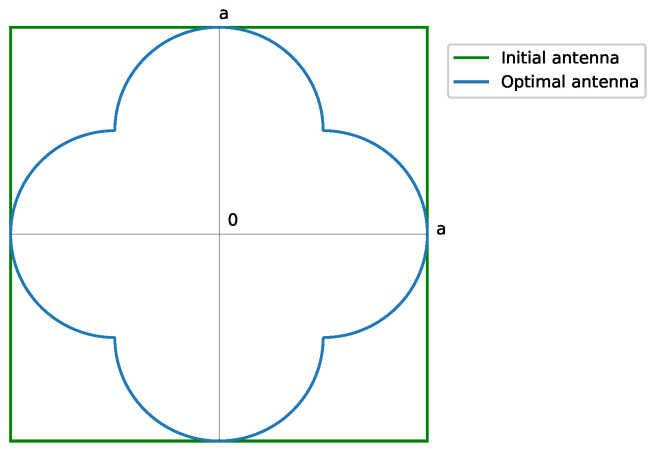
Shape of the required antenna for a rectangular footprint pattern of 1:1 aspect ratio. The required antenna matches the available one. The contour is obtained from Equation (Equation 10). The solid green line represents the initial antenna, while the solid blue line is the optimal antenna.

**Figure 3 sensors-23-04982-f003:**
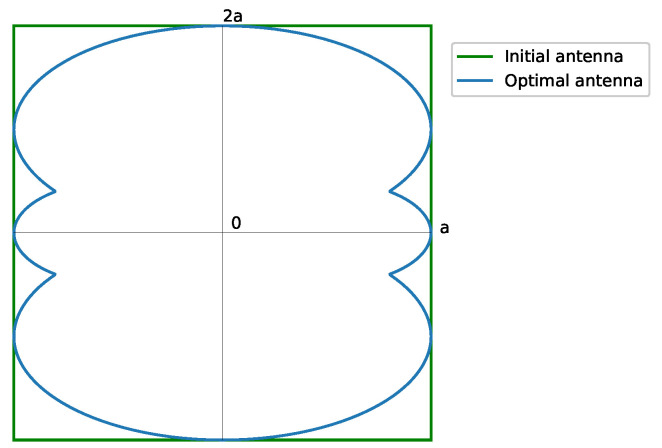
Shape of the required antenna for a rectangular footprint pattern of 2:1 aspect ratio. The required antenna matches the available one. The contour is obtained from Equation (Equation 10). The solid green line represents the initial antenna, while the solid blue line is the optimal antenna.

**Figure 4 sensors-23-04982-f004:**
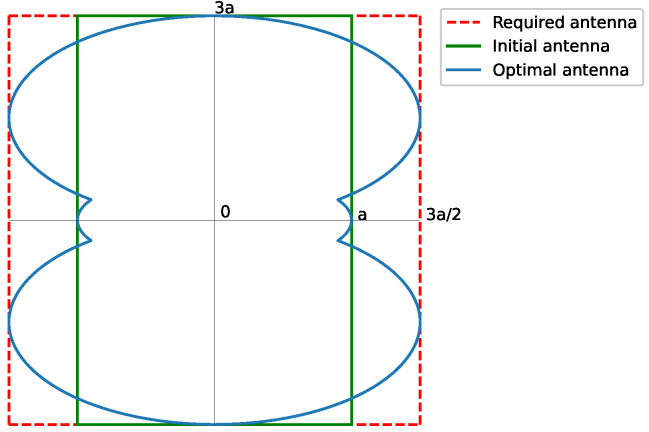
Shape of the required antenna for a rectangular footprint pattern of 3:1 aspect ratio. The required antenna exceeds the available one. The contour is obtained from Equation (Equation 10). The solid green line represents the initial antenna, while the dashed red line is the required antenna, and the solid blue line is the optimal antenna.

**Figure 5 sensors-23-04982-f005:**
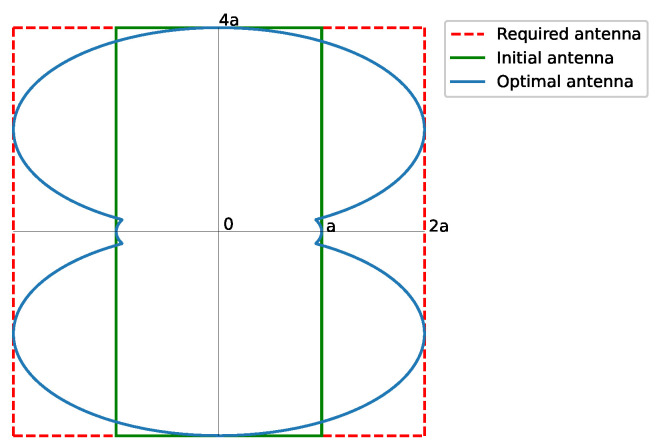
Shape of the required antenna for a rectangular footprint pattern of 4:1 aspect ratio. The required antenna exceeds the available one. The contour is obtained from Equation (Equation 10). The solid green line represents the initial antenna, while the dashed red line is the required antenna, and the solid blue line is the optimal antenna.

**Figure 6 sensors-23-04982-f006:**
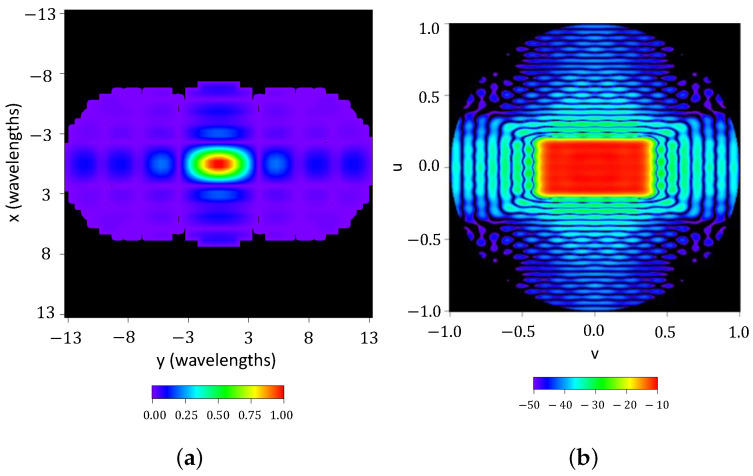
Normalized aperture distribution and interpolated image of the reconstructed pattern with a threshold level set at −50 dB from the antenna contour shown in Figure 3. (**a**) Normalized aperture distribution. (**b**) Reconstructed pattern.

**Figure 7 sensors-23-04982-f007:**
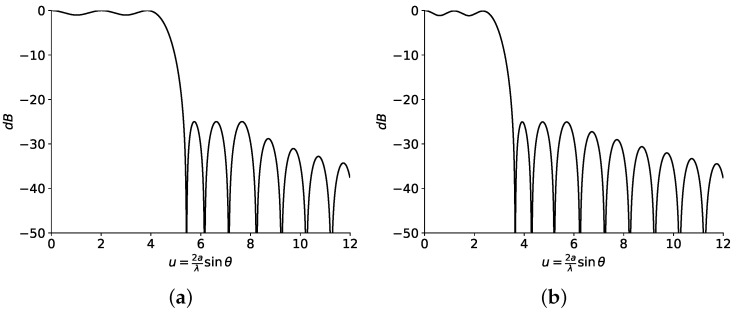
Patterns from (**a**) real and (**b**) complex roots. The HPBW for each set of roots is obtained at (**a**) $u_{0}=4.54$ and (**b**) $u_{0}=2.86$. Produced with a side-lobe level SLL=−25 dB, $\bar{n}=6$ inner roots, M=2 ripple cycles, and a ripple level of ±0.5 dB.

**Figure 8 sensors-23-04982-f008:**
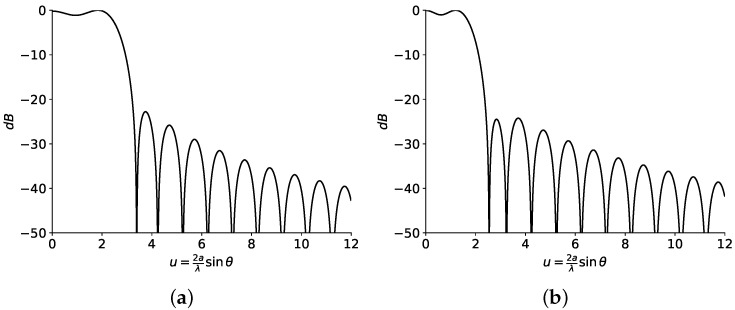
Patterns from (**a**) real and (**b**) complex roots. The HPBW for each set of roots is obtained at (**a**) $u_{0}=2.52$ and (**b**) $u_{0}=1.75$. Produced with a side-lobe level SLL=−25 dB, $\bar{n}=5$ inner roots, M=1 ripple cycles, and a ripple level of ±0.5 dB.

**Figure 9 sensors-23-04982-f009:**
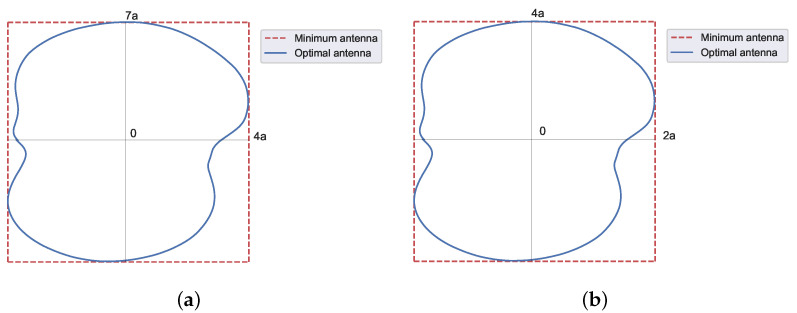
Shape of the required antenna for the case of the EuTELSAT antenna contour. The contour is obtained from Equation (Equation 9), using as initial pattern both (**a**) real and (**b**) complex flat-topped beam pattern boundaries with two filled zeros. The dashed red line represents the minimum antenna, while the solid blue line is the optimal antenna.

**Figure 10 sensors-23-04982-f010:**
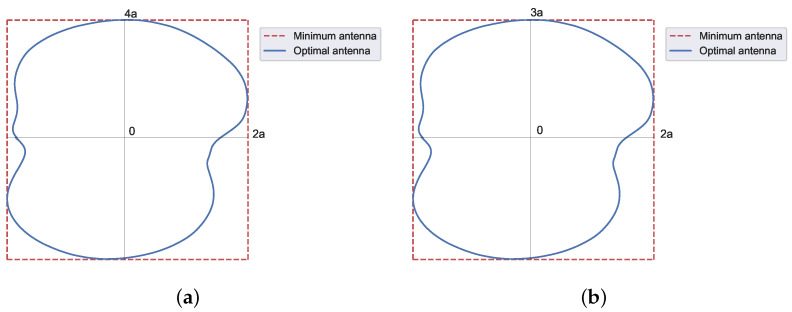
Shape of the required antenna for the case of the EuTELSAT antenna contour with one only zero. The contour is obtained from Equation (Equation 9), using as initial pattern both (**a**) real and (**b**) complex flat-topped beam pattern boundaries with one filled zero. The dashed red line represents the minimum antenna, while the solid blue line is the optimal antenna.

**Figure 11 sensors-23-04982-f011:**
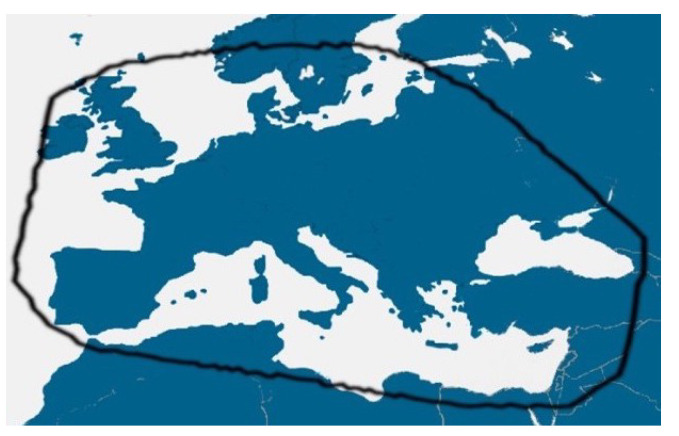
Contour of continental Europe covered by the EuTELSAT satellite.

## Data Availability

Data is contained within the article.

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
