# Peer review of "Maximizing Antenna Array Aperture Efficiency for Footprint Patterns"

_sensors, 2023, doi:10.3390/s23104982_

Round 1

Reviewer 1 Report

This paper studied a method for maximizing antenna array aperture efficiency for footprint patterns by optimizing the contour of the array. The derivation process is solid. I recommend accepting this paper after a little modification.

1. The abstract is recondite in some ways. The authors may rewrite the abstract to make it easier to follow.

2. The formulas in [Section Materials and Methods] are too many, which brings about an unsmooth reading experience. I think the formulas can be put into the appendix; only the main results should be shown in the main body.

Author Response

1. The abstract is recondite in some ways. The authors may rewrite the abstract to make it easier to follow

Reply: Thank you for the suggestion. We have rewritten the abstract, making the ideas more clear.

2. The formulas in [Section Materials and Methods] are too many, which brings about an unsmooth reading experience. I think the formulas can be put into the appendix; only the main results should be shown in the main body.

Reply: Thank you for your comment. Nevertheless, as we try to respect the predefined article structure, we do not think moving the formulas to the Appendix would help following the mathematical deductions within the article.

Reviewer 2 Report

COMMENTS:

The paper is very well written and presents a method to define the boundary of aperture antennas to synthesize footprint patterns and analyzes their aperture efficiencies. An interesting example of solutions to the asymmetric coverage defined by the European Telecommunications Satellite Organization (EuTELSAT) is described in the manuscript. It is a relevant result and is certainly of interest to most of the readers of Sensors. Some suggestions are listed below to give the reader more details regarding the solution discussed in the paper.

1) Line 28: “its contour has to be proportional to its half-power beamwidth (HPBW) in every phi-cut”, but the Abstract says “inversely proportional.”

2) The authors could add a comment regarding the polarization of the aperture distribution K0 first presented in Line 40.

3) The authors could add a reference for equations (4) and (5).

4) The authors could show the aperture distribution (graphical form) for at least one of the patterns synthesized in the paper.

5) The authors could also include a reference or mention techniques to generate the aperture distributions obtained in the examples shown in the paper, especially for the EuTELSAT, due to its practical use.

Author Response

1) Line 28: “its contour has to be proportional to its half-power beamwidth (HPBW) in every phi-cut”, but the Abstract says “inversely proportional.”

Reply: Thank you for the corrections, we have changed that in the main text.

2) The authors could add a comment regarding the polarization of the aperture distribution K0 first presented in Line 40.

Reply: A comment was added regarding the polarization of the aperture distribution K0.

3) The authors could add a reference for equations (4) and (5).

Reply: Thank you for the comment, a reference [19] has been added to equations 4, 5.

4) The authors could show the aperture distribution (graphical form) for at least one of the patterns synthesized in the paper.

Reply: Although it is not the aim of the present article to deeply study pattern synthesis, but contour antenna for maximizing aperture efficiency. Nevertheless, we realize its importance and we have included an example for a rectangular footprint pattern.

5) The authors could also include a reference or mention techniques to generate the aperture distributions obtained in the examples shown in the paper, especially for the EuTELSAT, due to its practical use.

Reply: The synthesized patterns use the technique developed in the reference [18] of the main text. Given the complexity of this case, we are currently working in
some new, more efficiency approaches, although it is out of the scope of the
present work.

Reviewer 3 Report

The authors discuss a procedure for determining the contour of an antenna array to realize a specific footprint pattern.

I have some comments for the authors that must be addressed to improve the paper before its possible publication.

a) The authors must clarify the novel contribution of the paper; it seems that most of the presented have been discussed in previous works of the same authors. 

b) The method described in section 2 is unclear; some quantities are used before being defined, and others are never explicitly explained. Using some figures/schemes could help the reader understand the employed procedures.

c) The description of the method done by the authors suggests the optimality of the proposed procedure, but this optimality is never fully demonstrated. The authors should clarify whenever the proposed approach is a "heuristic" and/or explain the limits of the method described in the paper.

d) Some patterns are provided to validate the procedure, but the corresponding excitations realizing those patterns are not given. Since there is no space limit, please provide them in the text of the paper or as additional material.

e) In the most interesting case discussed, the European coverage examples, no patterns are given: the authors should provide the array's excitations that realize the mentioned coverage as well as the effectively realized pattern; otherwise, the overall approach of the paper remains not validated.

Author Response

a) The authors must clarify the novel contribution of the paper; it seems that most of the presented have been discussed in previous works of the same authors. 

Reply: The novelty of the paper is highlighted within the main text, specially in the final part of the introduction. However, we have slightly reformulated the abstract
in order to clarify this.

b) The method described in section 2 is unclear; some quantities are used before being defined, and others are never explicitly explained. Using some figures/schemes could help the reader understand the employed procedures.

Reply: Thank you for the suggestion. However, we think all quantities are already
defined in the main text before using them. As well, we have followed the same
order used in previous works, such as in the reference [20] of the article, which
we think is more clear for novice readers.

c) The description of the method done by the authors suggests the optimality of the proposed procedure, but this optimality is never fully demonstrated. The authors should clarify whenever the proposed approach is a "heuristic" and/or explain the limits of the method described in the paper.

Reply: It is always guaranteed that the contour of the antenna presents the desired half-power beamwidth for each ϕ-cut, also maximizing the aperture efficiency, that is the main purpose of the present paper.

d) Some patterns are provided to validate the procedure, but the corresponding excitations realizing those patterns are not given. Since there is no space limit, please provide them in the text of the paper or as additional material.

Reply: Thank you for your comment. Although it is not the aim of the present article to deeply study pattern synthesis, but contour antenna for maximizing aperture efficiency. Nevertheless, we realize its importance and have included an example for rectangular footprint patterns.

e) In the most interesting case discussed, the European coverage examples, no patterns are given: the authors should provide the array's excitations that realize the mentioned coverage as well as the effectively realized pattern; otherwise, the overall approach of the paper remains not validated.

Reply: The synthesized patterns use the technique developed in the reference [18]
of the main text(). Given the complexity of this case, we are currently working
in some new, more efficieny approaches, although it is out of the scope of the
present work.

Round 2

Reviewer 3 Report

I am sorry to say that, but the authors have not answered the comments in the previous review satisfactorily.

a) No efforts have been made to clarify the novel contribution of the paper. Marginal modifications to the abstract are not sufficient.

b) The authors have not improved the description of the method, which remains obscure if the reader does not know the previous papers from the same authors.

c) The authors chose not to answer the question on the optimality of the approach: is the solution found the optimal one or a sub-optimal solution?

d) A single aperture distribution has been added (Fig.6a), but the plot is difficult to read. It would be better to use a logarithmic scale for amplitudes and an "image" plot, as in Fig.6b.

e) The lack of pattern and excitations for the European coverage examples leaves the innovative part of the paper not-validated since examples of the rectangular footprint antennas, similar to the ones depicted in this manuscript, were discussed in previous articles from the same authors.

Author Response

a) No efforts have been made to clarify the novel contribution of the paper. Marginal modifications to the abstract are not sufficient.

Reply: Thank you, the main contributions of this paper have been clarified in the
introduction.

b) The authors have not improved the description of the method, which remains obscure if the reader does not know the previous papers from the same authors.

Reply: We are sorry our corrections to the main text were not found to be clear enough. We have added information to some intermediate steps, which aim to complete the development of the method. We have included information regarding equations 5, 6, and more references to equation 7 (which is a compact representation of the real and complex flat-topped beam patterns, both used in the present work).

c) The authors chose not to answer the question on the optimality of the approach: is the solution found the optimal one or a sub-optimal solution?

Reply: Thank you for your answer, although we would like to highlight that this study is not focused on antenna pattern synthesis. However, in order to obtain optimal solutions, numerical optimizations must be performed. It is true that the
disposition of radiating elements within a, for example, rectangular grid, would
not exactly adapt the optimal antenna contour. Therefore, it is necessary to
optimize the disposition of these radiating elements. In this sense, starting from
the antenna contour proposed in this work, which is very close to the optimal
solution, allows getting optimal solutions with global or even local optimization
methods. A comment regarding this topic has been added to the main text.

d) A single aperture distribution has been added (Fig.6a), but the plot is difficult to read. It would be better to use a logarithmic scale for amplitudes and an "image" plot, as in Fig.6b.

Reply: Following your advice, we have substituted the 3D plot of the distribution of
intensities for an ’image’ plot.

e) The lack of pattern and excitations for the European coverage examples leaves the innovative part of the paper not-validated since examples of the rectangular footprint antennas, similar to the ones depicted in this manuscript, were discussed in previous articles from the same authors.

Reply: Thank you for the answer. However, the synthesis of the European coverage
example is currently being investigated by the authors, as developing the synthesis
with the aimed shape is not straightforward. Indeed, it has already been
studied in previous articles, including the reference [18] of the main text.